



# Earliest meteorological readings in San Fernando (Cádiz, Spain)

Nieves Bravo-Paredes[1], María Cruz Gallego[1], Ricardo M. Trigo[2,3] and José Manuel Vaquero[1]

[1]Departamento de Física, Universidad de Extremadura, Badajoz, Spain
[2]Instituto Dom Luiz (IDL), Faculdade de Ciências da Universidade de Lisboa, 1749-016 Lisbon, Portugal
[3]Departamento de Meteorologia, Universidade Federal do Rio de Janeiro, Rio de Janeiro, 21941-919, Brazil

*Correspondence to*: José Manuel Vaquero (jvaquero@unex.com)

**Abstract.** Cádiz and San Fernando are two nearby towns with a wealth of meteorological records due to their connection with the Spanish Royal Navy officers and enlightened merchants. Several previous works have already recovered a significant amount of meteorological records of interest in these localities. However, unnoticed previously more than 40,000 daily meteorological observations recorded at the Royal Observatory of the Spanish Navy (located in San Fernando) during the period 1799-1813 remained neither digitized nor studied. Here, we have carried out this important task describing the different steps undertaken to achieve it as well as the results obtained. The dataset is composed by different meteorological variables such as atmospheric pressure, air temperature, precipitation, or state of the sky. As a first step a quality control was carried out to find possible errors in the original data or in the digitization process. Moreover, the antique units were converted to modern units. Also, the metadata and an analysis of the data have been described. Finally, we study in detail the meteorological conditions in October 1805, during the Battle of Trafalgar and to check the possible local effects of the unknown volcanic eruption of 1809. The dataset is freely available to the scientific community and can be download at https://doi.org/10.5281/zenodo.7104289.

## 1 Introduction

Long-term series are very important to better understand the variability of the climate as well as to provide a more robust framework to validate long-term trends and extremes. In this context, early meteorological observations are critical to extend instrumental series, particularly during the 18th and 19th centuries. Therefore, climate data rescue should be a continuous effort part of a long term activity (Brönnimann et al., 2018). Recovering early meteorological observations represents a huge effort played by a large number of scientists all over the world. Some important examples of structured international efforts

in this line of activity are the Atmospheric Circulation Reconstructions over the Earth (ACRE) (http://www.met-acre.org/) and the International Data Rescue (I-DARE) Portal (https://www.idare-portal.org/). Early meteorological data have been also recovered in Latin-America and the Caribbean (Domínguez-Castro et al., 2017), in Canada (Slonosky, 2003), in South Africa (Picas et al., 2019), in Brazil (Farrona et al., 2012), and in Japan (Zaiki et al., 2006).

In Europe, different initiatives have been carried out to retrieve early meteorological data. For example, the ADVICE project

reconstructed and analyzed MSLP data from the 16th century onwards (Jones et al., 1999; Luterbacher et al., 2000). There





have been more initiatives such as IMPROVE (Camuffo and Jones, 2002), HISTALP (Auer et al., 2007) or ERACLIM (http://www.era-clim.eu/) projects. More recently initiatives include, for example, the CHIMES project that compiles pre-national weather service observations in Switzerland and makes them available in digital format (Pfister et al., 2019; Brugnara et al., 2020).

In the Iberian Peninsula (hereafter IP), different initiatives in the last two decades can be found. For example, Alcoforado et al. (2012) retrieved early Portuguese meteorological measurements from 18th century, while project SIGN focused in 19th and early 20th century data in Portugal (Morozova and Valente, 2012; Bližňák et al., 2015). Domínguez-Castro et al. (2014) recovered more than 100000 early meteorological observations prior to 1850 in Spain, and Vaquero et al. (2021) recovered more than 750000 instrumental data from 1826 to mid-20th century in the Extremadura region (interior SW Iberia) presented

in the CliPastExtrem database. In the Bay of Cádiz, there are several initiatives to rescue meteorological data that led to a wide number of publications (Barriendos et al., 2002; Gallego et al., 2007; Rodrigo, 2012; Domínguez-Castro et al., 2014; Wheeler, 1995; Rodrigo, 2019).

Figure 1 summarizes the data retrieved for the cities of Cádiz (references in black) and San Fernando (references in red). The first instrumental data were recovered by Wheeler (1995) using temperature measurements made during astronomical

observations (passage of the Sun through the Cadiz meridian) in 1776. Barriendos et al. (2002) studied the long records of the Urrutia brothers in Cádiz. Gallego et al. (2007) recovered the meteorological records of the "*Torre de Vigía*" at Cádiz. The Royal Observatory of the Spanish Navy (ROA, from the Spanish "*Real Instituto y Observatorio de la Armada*") (located in San Fernando) has published its meteorological records in annual volumes since 1870. In addition, other short series have been recovered (Rodrigo, 2012, 2019; Domínguez-Castro et al., 2014).

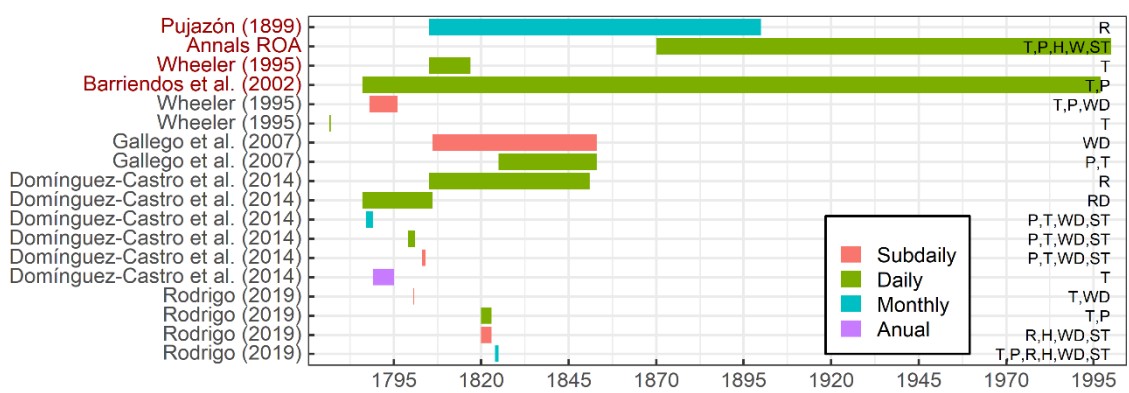


**Figure 1: Data retrieved for the cities of Cádiz (in black) and San Fernando (in red). The meteorological variables of each dataset are on the right: T – temperature, P – pressure, R – rain, WD – wind direction, H – humidity, RN – rainy days, W – different wind variables, ST – state of the sky.**

The main aim of this work is to retrieve the early meteorological observations measured in San Fernando (Cádiz) for the

period 1799-1813. Moreover, a quality control and unit changes are carried out to make the data usable. Finally, an analysis of the data is performed. The structure of the paper is as follows: Section 2 describes the data and the instruments used; the



method used for the quality control and the unit change of the data is presented in Section 3; the corresponding results are presented in Section 4; data applications are shown in Section 5; finally, conclusions of this work are presented in Section 6.

## 2 Data and instruments

The dataset of early meteorological observations here recovered were recorded in the ROA during the period 1799-1813. The ROA is located in the city of San Fernando (Cádiz) in the southwest of the IP (36° 27′ 42″ N, 6° 12′ 20″ W). Figure 2(a) shows the location of the city of San Fernando in the IP. Figure 2(b) shows the location of San Fernando in the Bay of Cádiz. Figure 2(c) shows the location of the ROA in the city of San Fernando.

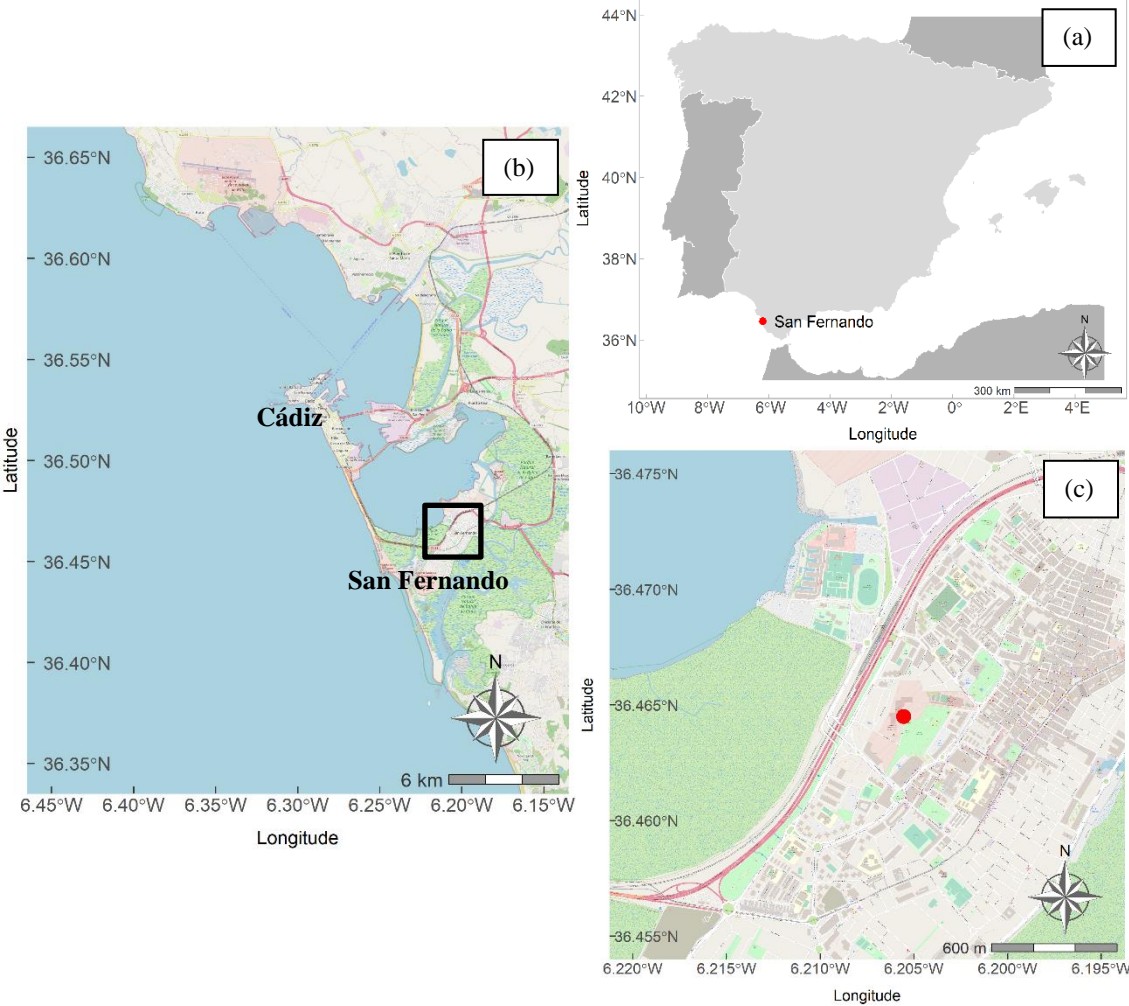

**Figure 2: (a) The red point shows the location of the city of San Fernando in the IP; (b) The location of San Fernando in the Bay of Cádiz. The box shows the region enlarged in (c); (c) The red point shows the location of the Royal Observatory of the Spanish Navy (ROA) in the city of San Fernando. The maps were created from Instituto Geográfico Nacional (www.ign.es) and the *ggspatial* package in R (Dunnington, 2020).**





The dataset of the ROA here presented (hereafter SF1799-1813) corresponded to a plan to carry out astronomical

observations at the Observatory starting in 1798. The plan was divided in four classes: *tiempo*, *planetas*, *longitudes terrestres* and *física celeste*. Specifically, the observations of SF1799-1813 corresponded to the fourth class (*Física Celeste*) (Lafuente and Sellés, 1988). Original data were preserved at the *Real Instituto y Observatorio de la Armada* in San Fernando in a logbook (catalog number: AH1080).

These meteorological observations were recorded with different instruments and were measured in different units. In total,

45862 meteorological observations of the ROA dataset were recorded during the period 1799-1813. Figure 3 shows the observation period covered for each meteorological variable of the SF1799-1813 dataset. Pink, green and blue bars correspond to sub-daily observations: 6am, 12am, 6pm and 12pm hours in pink; 8am and 2pm hours in green; 2pm hour in blue. Gray bars correspond to monthly observations. Unfortunately, there are two large gaps in the periods 1802-1804 and 1810-1812. The cause of the first gap could be that the work of the officers carried out on the sea was more valued than the

scientific work in the ROA. Because of this, there were few officers working at ROA. Unfortunately, the absence of the officers often provokes that meteorological observations were not made (Lafuente and Sellés, 1988). The second gap is related to the movement of the observer to the Torre de Zimbrelos. The observer was assigned to watch over the French due to Napoleon's occupation during the Spanish War of Independence or Peninsular War.

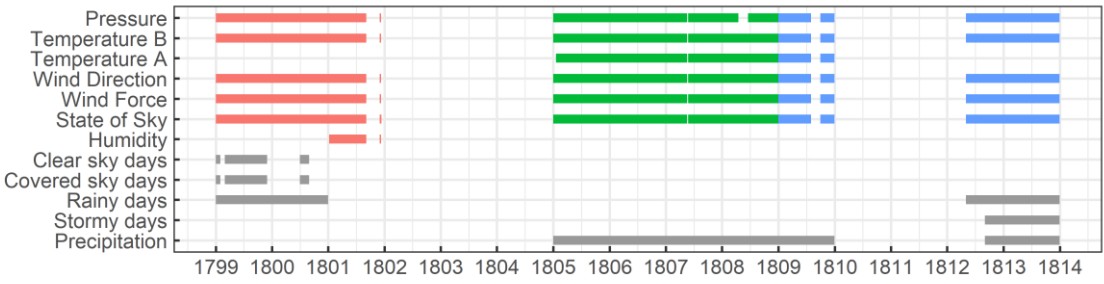

**Figure 3: Observational period covered by the new dataset for each meteorological variable. Pink, green and blue bars correspond to sub-daily observations: 6am, 12am, 6pm and 12pm in pink; 8am and 2pm in green; 2pm in blue. Gray bars correspond to monthly observations.**

Additionally, another dataset of the ROA is used for comparison purposes with the SF1799-1813 dataset. This dataset covers the recent period 1997-2021 and can be found at https://datosclima.es/ (hereafter SF1997-2021). Finally, a monthly global

paleo-reanalysis (EKF400v2) generated to cover the 1600 and 2005 time period by Valler et al. (2021) was used to make comparisons with the SF1799-1813 dataset.

The station elevation appears in the metadata but is different for each period. There is not station elevation data in the metadata for the period 1799-1801. For the period 1805-1809, the station elevation was recorded as 142.5 Burgos foot (1 Burgos foot = 0.2786 m). For the period 1812-1813, the elevation was recorded as 100 feet from sea level. This last value

has been chosen for the period 1799-1801 due to the observations were registered with the same instruments and could be the same location. A brief description of each variable is given below:





- **Pressure:**

Pressure observations were recorded with two different mercury barometers and in different units:

- 1799-1801 and 1812-1813: pressure observations were recorded subdaily in English inches with a George Adams
barometer.
- 1805-1809: pressure observations were recorded subdaily in French inches with a Megnié barometer.

- **Temperature (thermometer attached to the barometer):**

Mercury barometers typically had a thermometer attached to do the temperature correction for pressure observations. Temperature observations of the thermometers attached to the barometers were recorded subdaily with different
thermometers in different units:

- 1799-1801 and 1812-1813: the thermometer attached to the George Adams barometer registered subdaily observations in Fahrenheit degrees.
- 1805-1809: the thermometer attached to the Megnié barometer recorded subdaily observations in Reaumur degrees.

- **Air temperature:**
Air temperature was recorded with a Dollond thermometer in Fahrenheit degrees for the period 1805-1809.

- **Wind direction:**

Wind direction observations were recorded subdaily by following the wind rose directions. The wind rose was divided into 32 directions: N, NpNE, NNE, NEpN, NE, NEpE, ENE, EpNE, E, EpSE, ESE, SEpE, SE, SEpS, SSE, SpSE, S, SpSW, SSW, SWpS, SW, SWpW, WSW, WpSW, W, WpNW, WNW, NWpW, NW, NWpN, NNW, NpNW. The term "p" between
directions means "*por"* in Spanish ("by" in English). In the original data, these wind directions were recorded as, for example, "NE1/4E". In the digitization process, the term "NE1/4E" was replaced by "NEpE".

The prevailing wind direction in the 24h were recorded for the period 1812-1813.

- **Wind force:**

Wind force observations were recorded subdaily in textual form using descriptors. Calvo et al. (2008) wrote a dictionary to
clarify the archaic wind force terms in a comprehensible format. This dictionary was used to check the wind force terms from the original document. Most of the terms for the whole period appear in the dictionary.

The prevailing wind force in the 24h were recorded for the period 1812-1813.

- **State of the sky:**

The observers gave a detailed description of the state of the sky subdaily during the period 1799-1801. Clear or covered sky,
amount of clouds, rain or other type of precipitation are examples of what usually appears in the description.

The state of sky was recorded differently since 1805. The description was summarized in just one word, for example, "clear" to indicate that the sky was cloudless.

The prevailing state of the sky in the 24h were recorded for the period 1812-1813.





Moreover, the number of clear, covered, rainy and stormy days were also registered monthly in different periods. Clear,
covered and rainy days were recorded for the period 1799-1800. Rainy and stormy days were recorded for the period 1812-
1813.

- **Humidity:**

There is not information about the humidity in the metadata. The humidity was registered only for the period 06/01/1801-
11/12/1801.

- **Precipitation:**

The precipitation was recorded weekly in Burgos inches (1 inch = 23.22 mm) for the period 1805-1808 and monthly for the
year 1809 and the period 1812-1813.

## 3 Method

Data were digitized by key entry from photographs taken from the manuscript. The dataset was analyzed through a basic
quality control applied to detect possible errors in the digitization process or suspicious values. Moreover, the units were
changed to modern SI units. The steps will be described in the following sections.

### 3.1 Quality control

Digitized data often contains errors as a result of typing an incorrect number in the digital document instead of the real
number of the original document. These errors should be detected and corrected whenever possible. Moreover, quality
control could detect suspicious values. These values were digitized correctly. Therefore, it is possible that the observers
wrote an incorrect number instead of the correct number (e.g.: a maximum daily temperature with lower values than the
corresponding minimum daily temperature). Suspicious values detected by quality control are flagged with an asterisk and
are not corrected.

A basic quality control was carried out in order to detect possible typos, errors or suspicious values in the digitization
process. The quality control consists of the following:

1. **Visual check:** some variables are bounded by upper and/or lower limit values. For example, pressure lines cannot
exceed the value of 12. If there are values that exceed this threshold, the values are checked against the original data
and corrected.

2. **Compare means:** the daily and monthly means calculated by the observers were digitized and compared with the
means computed from the digitized data. If the means are not equal, the values corresponding to that mean are
checked against the original data and corrected whenever a mistake is detected.

3. **Extreme values:** in order to detect outlier values, the mean plus/minus three standard deviations are calculated by
the authors. Values that exceed this threshold are checked against the original data and corrected if there is an
obvious error.



4. **Consecutive values:** the difference of the values between consecutive days is also calculated. This difference should not exceed a predefined threshold. The threshold is estimated by the mean plus/minus three standard deviations of the differences. Values above this threshold could be due to an error, a suspicious value or an outlier. Values that exceed this threshold are checked against the original data and corrected if it is an error.

The four quality control steps mention above were applied to pressure and temperature variables (temperature related to the

thermometer attached to the barometer). For the air temperature variable and for the humidity only steps 1, 3 and 4 were applied.

### 3.2 Unit change

### 3.2.1 Temperature

Temperature was measured using the Reaumur and Fahrenheit scales in different periods. Temperatures were converted to

Celsius using the two following equations where Eq. (1) converts Reaumur to Celsius, while Eq. (2) converts Fahrenheit to Celsius.

$$T = \frac{5}{4} T_R \tag{1}$$

$$T = \frac{5}{9} (T_F - 32) \tag{2}$$

### 3.2.2 Pressure

Pressure was measured in two different length units, English and French inches, in two different periods. Pressure measurements originally registered in inches were converted to meters. Equation (3) is used to convert the English inch to meters. Equation (4) is used to convert the French inch to meters.

$$h = 0.0254 h_{mercury} \tag{3}$$

$$h = 0.02707 h_{mercury} \tag{4}$$

• **Reduction to 0 °C:**

Mercury expands and shrinks depending on the temperature. Therefore, meteorological observations recorded with a mercury barometer should be reduced to 0 °C. The density of the mercury changes with temperature. The following equation gives the density of mercury as a function of temperature:

$$\rho_{Hg} = \frac{\rho_0}{1 + \alpha T}$$

where $\alpha = 0.0001818$ K$^{-1}$ is the volumetric thermal coefficient for mercury at 0 °C, $\rho_0 = 13595.1$ Kg m$^{-3}$ is the density of mercury at 0 °C and $T$ is the temperature of the thermometer attached to the barometer.



The barometer scale could expand due to temperature. Therefore, the scale must be corrected. The linear thermal expansion equation can be written as:

$$\beta = \frac{1}{L}\left(\frac{\Delta L}{\Delta T}\right)$$

Thus, expanding on this equation, the height of the scale corrected to 0 °C ($h_0$) can be calculated with Eq. **¡Error! No se encuentra el origen de la referencia.**).

$$h_0 = [1 + \beta T]h \tag{5}$$

Finally, the pressure in meters of mercury is converted to Pascal through Eq. (6):

$$p_0 = g_n \frac{\rho_0}{1+\alpha T}[1 + \beta T]h \tag{6}$$

where $h$ is the height of the mercury, $T$ is the temperature of the thermometer attached to the barometer, and $g_n$=9.80665 m s$^{-2}$ is the normalized gravity.

- **Correction for local gravity:**

Equation **¡Error! No se encuentra el origen de la referencia.**) gives the local gravity as a function of the latitude ($\varphi$) and the station elevation ($H$).

$$g_{\varphi,H} = 9.8062(1 - 0.0026442\cos 2\varphi - 0.0000058\cos^2 2\varphi) - 0.000003086H \tag{7}$$

Finally, Eq. (8) gives the pressure corrected to 0 °C and local gravity.

$$p_n = p_0 \frac{g_{\varphi,H}}{g_n} \tag{8}$$

- **Reduction to mean sea-level pressure:**

Pressure values at different altitudes should be reduced to mean sea-level pressure when comparisons between them are
carried out (WMO, 2018). The equation to obtain pressure values reduced to MSL is:

$$p_{MSL} = p_n \cdot \exp\left(\frac{\frac{g_n}{R}H}{T+a\frac{H}{2}}\right) \tag{9}$$

where $H$ is the station elevation, $g_n$ is the normalized gravity, $R$=287.05 J Kg$^{-1}$ K$^{-1}$ is the gas constant of dry air, $T$ is the air temperature, and $a$=0.0065 K m$^{-1}$ is the assumed lapse-rate in the fictitious air column extending from sea level to the station elevation level.

Air temperature observations are only available for the period 1805-1809. The coefficient of determination between air temperature observations and temperatures of thermometer attached to the barometers observations for the same period is





$R^2$=0.8638. Thus, $T$ in Eq. (9) is considered as the temperature of the thermometer attached to the barometer to cover the entire period 1799-1813.

## 4 Results and discussion

In total, 19618 values were analyzed by the quality control procedures and circa 300 digitized values were detected as errors in the digitization process and were corrected. Three values were detected as suspicious and were flagged with an asterisk. This represents 1.54 % of the total data analyzed (0.66 % of the total data). The total errors in each variable vary substantially: 82 errors were detected in pressure observations, 183 in temperature observations (related to the thermometer attached to the barometer), 18 in air temperature and 17 in humidity observations. The number of total readings analyzed for

pressure were 7652, for temperature (thermometer attached to the barometer) 7774, for air temperature 3176 and for humidity 1016. Therefore, the percentage of errors detected in each variable is, respectively: 1.07 % for pressure, 2.35 % for temperature (related to the barometer), 0.57 % for air temperature and 1.67 % for humidity.

Both pressure series (SF1799-1813 and SF1997-2021) have been compared. For pressure data, the available period of the SF1997-2021 dataset is 2006-2021. Pressure values of SF1799-1813 are lower than the more recent pressure values of 2006-

2021. We have made an adjustment of the SF1799-1813 values to be able to compare them with modern data. In fact, two adjustments are required in different periods due to the two different barometers used (George Adams and Megnié barometers). For the period 1799-1801 and 1808-1813, a value of 10.3 hPa was added to the original observations and for the period 1805-1809 a value of 4.9 hPa was added. Other studies found this same problem and have applied similar correction adjustments. For example, the pressure series for Leiden required an adjustment of 16.7 hPa (Können and

Brandsma, 2005). An adjustment between 9.0 hPa and 10.5 hPa was equally needed for the pressure series of London (Slonosky et al., 2001). Rodrigo (2019) found a difference between the monthly data of Cádiz for the period 1820-1822 and the mean values for the period 1901-1930 of roughly 3 hPa. According to these authors possible explanations for this problem could be related to the calibration of the instrument, imperfections in the barometers used, the existence of trapped air, or the unknown correct altitude of the barometer.

Time series plots of the monthly pressure means computed from the daily SF1799-1813 and SF1997-2021 series are shown in Fig. 4. Dashed gray line represents the pressure means of SF1799-1813 series for the period 1799-1813. Solid gray line shows the adjusted pressure observations of SF1799-1813, at daily (upper panel, Fig. 4) and monthly (lower panel, Fig. 4) scales. Black line represents the monthly average pressure of the SF1997-2021 series for the period 2006-2021. Dashed black line shows the monthly pressure of EKF400v2 dataset for the period 1799-1813. This curve corresponds to the nearest

grid point of the regular grid of the EKF400v2 dataset.

Note that there is a monthly decrease in pressure values in the first months of 1808 (Fig. 4). Cornes et al. (2012) recovered pressure values measured with a Megnié barometer and found that the vacuum of Megnié barometer suffered a drift of a rate of 0.02 hPa per month in the Paris Observatory. It is possible that the Megnié barometer of the ROA (SF1799-1813) suffered





a similar drift as the Paris Observatory. In the metadata, the observer wrote in the manuscript that the readings of the Megnié
barometer in 1808 were 0.22 inches of pressure lower than the George Adams barometer. Moreover, the readings were lower
than when the barometer was installed. In 1809, the observer wrote in the manuscript that the barometer was repaired in
August. The difference of 0.22 inches of pressure between the Megnié and the George Adams barometer mentioned by the
observer is checked here. For the period 1808-1809, the pressure observations are 0.22 inches of pressure lower than the
pressure observations for the period 1799-1801. Due to this fact, it has been added 0.22 inches of pressure to the original
pressure observations (already corrected in Fig. 4) for the period 1808-1809.



**Figure 4: Dashed gray line represents the pressure means computed from the daily SF1799-1813 series for the period 1799-1813. Solid gray line shows the adjusted pressure observations of SF1799-1813, at daily (upper panel) and monthly (lower panel) scales.**





**Black line represents the monthly pressure average cycle of SF1997-2021 series for the period 2006-2021. Dashed black line shows**
**the monthly pressure of EKF400v2 dataset for the period 1799-1813.**

In addition, the pressure values of the EKF400v2 dataset are higher than the pressure values of the SF1799-1813 and SF1997-2021 datasets (see Fig. 4, lower panel). The coordinates of the EKF400v2 dataset are close to those of the city of San Fernando but they are not the same. A possible explanation of the difference in pressure values between the different series can be found in this fact. However, the behavior of the EKF400v2 dataset is similar to the behavior of the SF1799-
1813 and SF1997-2021 datasets.

Figure 5 shows the daily air temperature of SF1799-1813 for the period 1805-1809 in gray. Daily average of the air temperature of SF1997-2021 for the period 1997-2021 is showed in solid black line. Dashed black line represents the monthly air temperature of EKF400v2 dataset for the period 1805-1809. Just like pressure data, this curve corresponds to the nearest grid point of the regular grid tan of the EKF400v2 dataset. As can be seen, the three curves (SF1799-1813, SF1997-

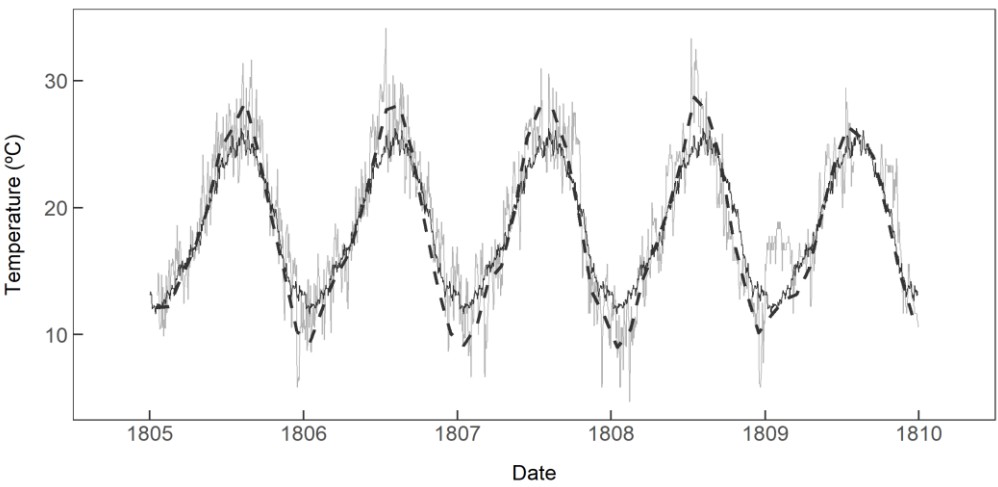


**Figure 5: Gray line represents the daily air temperature of SF1799-1813 for the period 1805-1809. Solid black line represents the mean monthly air temperature of SF1997-2021 for the period 1997-2021. Dashed black line shows the monthly air temperature of EKF400v2 dataset for the period 1805-1809.**

2021 and EKF400v2) show a similar behavior and there is an excellent compatibility between the air temperature for the
270 SF1799-1813 and SF1997-2021 datasets. The coefficient of determination between these both curves is $R^2$=0.9104. For the case of SF1799-1813 and EKF400v2 datasets, the coefficient of determination is $R^2$=0.9384. For the case of the deseasonalized values of SF1799-1813 and EKF400v2 datasets, the coefficient of determination is $R^2$=0.4119. The air temperature of SF1799-1813 dataset for the period 1805-1809 shows a seasonal cycle similar to the monthly average of the SF1997-2021 dataset and to the monthly temperature of EKF400v2 dataset. The temperature of SF1799-1813 in summer
275 months is slightly higher than the monthly average of the SF1997-2021 and is quite similar to the monthly temperature of EKF400v2 dataset. As can be seen in Fig. 5, the temperature of the three series in the first months of each year of the period





represented is quite similar except at the beginning of 1809. The temperature values for the first months of 1809 of SF1799-



**Figure 6: Frequency of the wind direction represented in wind compasses for the 1799-1813 period for each month and in total (upper).**

280





1813 dataset are considerably higher than the monthly temperature of the SF1997-2021 and EKF400v2 datasets for these months. Interestingly, similarly unusual high values appear in the temperature series of the thermometer attached to the barometer (not shown). In order to understand the robustness of this signal we checked other instrumental data series (not shown) to evaluate if this behavior is unique to San Fernando (and therefore a potential error) or if it occurs in other regions

of the world. Six different temperature series from Switzerland (Brugnara et al., 2020) present the same behavior for the beginning of 1809. Also, two temperature series from Madrid and Mallorca (Spain) (Domínguez-Castro et al., 2014) show a similar behavior. Therefore, it seems that this higher-than usual winter temperatures did occurred throughout Europe (at least) and possible even at the global scale.

Individual subplots represented in Fig. 6 show the frequency of the wind direction represented in wind compasses for the

1799-1813 period for each month and in total (upper subplot). In general, the compasses show five prevalent wind directions (NW, W, SW, SE, and ESE) varying with the seasonal cycle. In winter months (December, January and February), the predominant wind directions are N, NW and SW. The SE and ESE wind directions are also relevant but to a lesser extent. In spring months (March, April and May), the main wind directions are NW, SW and SE, while in summer months (June, July and August), the prevalent wind directions are W, SW and ESE. Finally, in autumn months (September, October and

November), the predominant wind direction is SW, but to a lesser extent other relevant wind directions are NW, SE and ESE. The predominant wind directions are in accordance to the location of San Fernando, which is closed to the Strait of Gibraltar. In general, the frequency of westerlies is higher than the easterlies (Sousa, 1987).

Figure 7 shows the number of observations and the corresponding percentage of the wind force per class (at the top) and the state of the sky (at the bottom) for the period 1799-1813 and 1805-1813, respectively. The terms used to categorize the wind

forces are presented with the original Spanish name. These terms could be converted to the Beaufort scale using CLIWOC Dictionary (García-Herrera et al., 2004). The predominant wind forces are three: *flojo* (Beaufort scale = 3), fresco (Beaufort scale = 6) and *fresquito* (Beaufort scale = 5). There are also two additional terms with a high number of observations: *bonancible* (Beaufort scale = 4) and *recio* (Beaufort scale = 7). Calvo et al. (2008) obtained similar results for Cádiz (about 10 km northwest of San Fernando) for the longer period of 1806-1852. The predominant state of the sky is *claro* (clear sky).

The states of sky *nubes* (cloudy) and *lluvia* (rainy) are also relevant. The term *otros* in the two figures of Fig. 6 represents those observations with more than one wind force or state of sky. Nevertheless, this term represents a low number of observations and is not significant.

The accumulated precipitation per month for the period 1805-1813 can be observed in Fig. 8. In the summer months there is not precipitation data. In the metadata there is not information about the absence of observations or the precipitation. For this

reason, daily pressure data and periods without rainfall have been plotted together (not shown) to check that there are not drops in pressure values when there is not precipitation data. According to the represented, it seems that the precipitation data are consistent. Monthly values above 100 mm occur almost every autumn-winter and the highest precipitation values were registered in the month of December in 1812 and October in 1813. Calvo et al. (2008) showed the absolute frequency





of precipitation (days per year) of Cadiz for the same period and obtained similar results. The maximum absolute frequency was recorded in 1813, with a frequency of 62 days per year. There is no data for the year 1812 (Calvo et al., 2008).

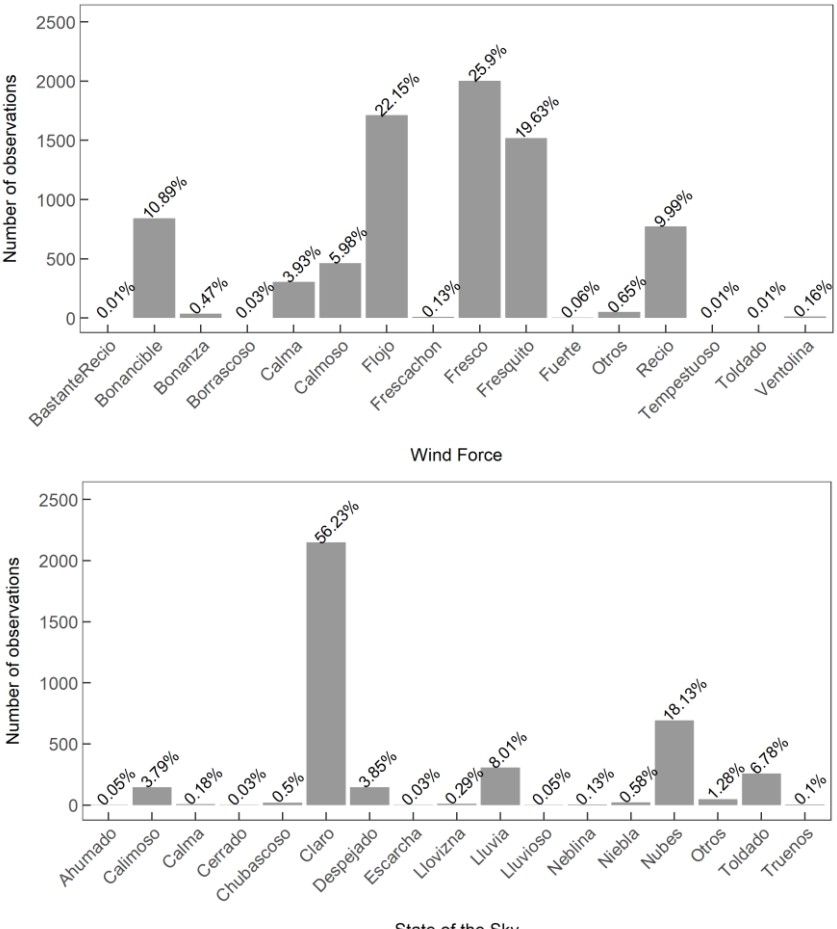

**Figure 7: Top) Number of observations and the corresponding percentage of the wind force per class for the period 1799-1813 at the top. Bottom) number of observations and the percentage of the state of the sky per class for the period 1805-1813.**

The values of sea level pressure (SLP) and the total precipitation (rain and snow) anomalies of the EKF400v2 dataset are

represented below for the months of November and December 1812 and the months of October to December 1813 (Fig. 9). The highest precipitation values were recorded in these months for the period 1805-1813 (see Fig. 8). In the months of February and March 1813 no precipitation was recorded, but the metadata shows several days of rain for February and one for March. Therefore, there could be some error in these months. The precipitation anomalies estimated for the month of December 1812 for southern Spain is around 200 kg m$^{-2}$ (Fig. 9). For the year 1813, high precipitation values were also

estimated for the months of October and December and virtually zero anomaly for the month of November for southern Spain. Pressure values of around 1020 hPa are estimated for the months of November and December 1813 and around 1018



hPa for October 1813. Overall, the large-scale atmospheric dynamics at the monthly scale is in good agreement with the precipitation values represented in Fig. 8 for these months.

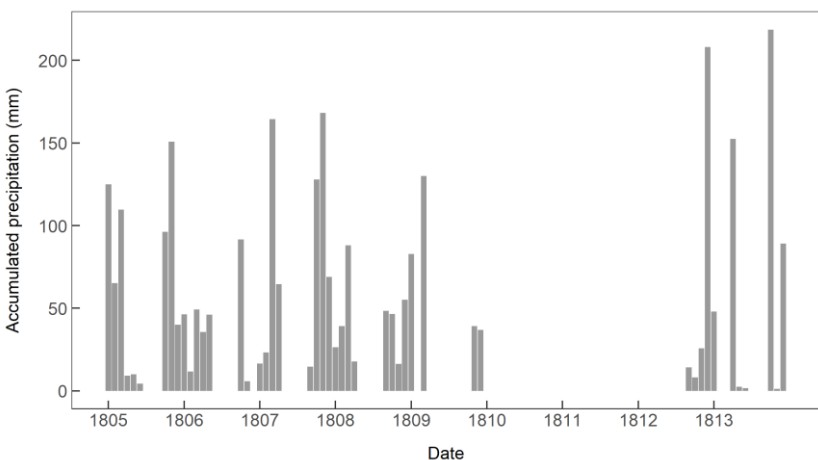

**Figure 8: Accumulated precipitation per month for the period 1805-1813.**

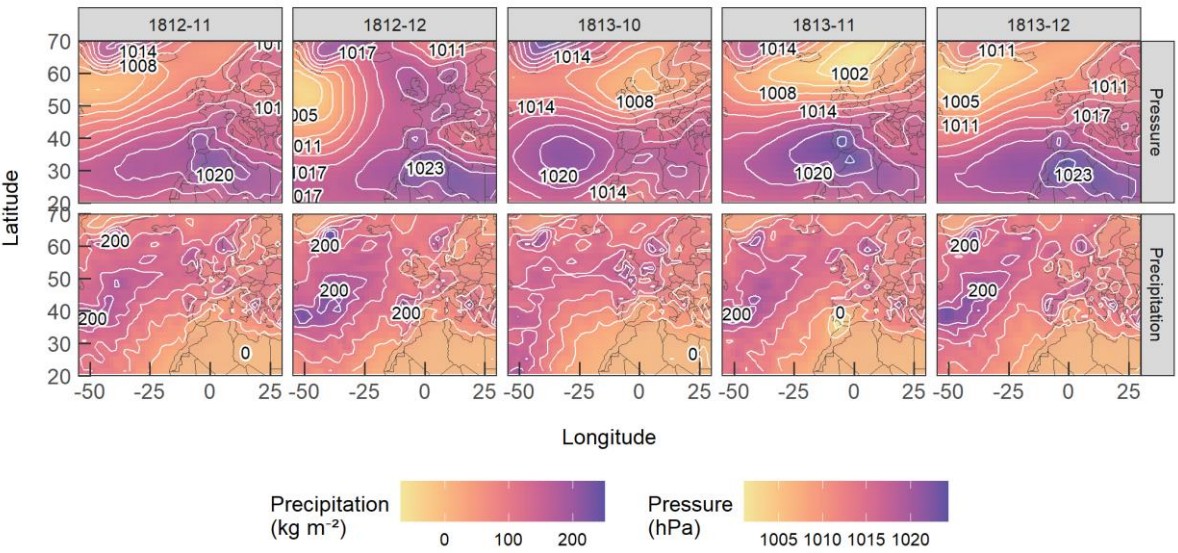

**Figure 9: Monthly SLP and monthly precipitation anomalies for the months that registered the maximum values of precipitation (see Fig. 7).**

**5 Data applications**

Once the observations of SF1799-1813 dataset have been digitized, analyzed and checked for different types of errors, it becomes possible to consider its use for different applications, namely to identify and characterize weather and climate events. Here, the data was employed to analyze in further detail the meteorological conditions during the important historical



event of the Battle of Trafalgar (October 21st, 1805) and also to check -on a longer time-framework- the effects on San Fernando of the unknown (but highly cited) volcanic eruption of 1809.

Wheeler (1985) studied the weather conditions during the Battle of Trafalgar. Pressure observations recorded in England were used in the study due to the lack of data near Trafalgar Cape. By coincidence, the city of San Fernando is located very close (circa 35 km) from Trafalgar Cape. Thus, weather conditions for October 1805 will be analyzed using the SF1799-1813 dataset. Daily pressure anomalies were calculated for the year 1805. Anomalies were calculated using pressure observations from Cádiz for the period 1955-2021 (the data can be downloaded at https://www.ecad.eu). The anomalies were

computed as the difference between the daily pressure values of SF1799-1813 dataset and the corresponding daily average of the Cádiz pressure observations. Mean plus/minus two and three standard deviations were calculated for the pressure observations of the Cádiz data. The first large pressure drop since the end of March 1805 was registered on October 21st, 1805. The pressure anomaly associated with this pressure drop exceeds the limit of three standard deviations. One of the synoptic scenarios proposed by Wheeler (1985) to explain the weather during the battle was through the presence of a cut-off

low system. Figure 10 represents the SLP and the geopotential height at 500 and 100 hPa of the EKF400v2 dataset for the months of September to November 1805. The cut-off lows present an isolated closed contour in the geopotential height maps in the middle and upper troposphere. As can be seen in Fig. 10, there are no isolated closed contours for the 500 and 100 hPa. In the SLP plots, the Azores High is weakened in October and the SLP near Trafalgar Cape is lower than during the other months. However, it is necessary to stress that the typical duration of the cut-off lows is only a few days (Nieto et al.,

2007), so it is very difficult to analyze their presence using monthly data alone. Therefore, it cannot be confirmed or dismissed at this stage if the weather during the Battle of Trafalgar was caused by a cut-off low system. In any case, the pressure dropped during the battle and the accumulated precipitation recorded during the fourth week of October in the SF1799-1813 dataset was 60.18 mm. These observations are compatible with the passage of a low-pressure system, which is common in the climatology of this geographical area, this being the most plausible meteorological scenario with the

currently available data. The mean accumulated precipitation for the month of October in San Fernando of the SF1997-2021 dataset is 60.8 mm. Thus, the precipitation value of 60.18 mm at the weekly scale in October 1805, indicates that it was a strong wet event.

The date of the volcanic eruption of 1809 is unknown, but detailed analysis of Antarctic and Greenland ice core records confirmed the existence of an important eruption in February 1809 (± 4 months) (Cole-Dai, 2010). This is in agreement with

other studies (Sigl et al., 2013, 2015). Also, a possible eruption in late November or early December (December 4th, 1808 ± 7 days) is suggested by Guevara-Murua et al. (2014) when analyzing atmospheric phenomena related to the stratospheric aerosols. The temperature for the year 1809 of the SF1799-1813 dataset shows a decreased range with higher (lower) than-average temperatures during winter (summer) months (as can be seen in Fig. 5). This decrease range could be partially caused by the unknown volcanic eruption of 1809 that likely induced cooler summer temperatures. Temperature anomalies

are calculated for the two thermometers available for the SF1799-1813 dataset (thermometer attached to the barometer and air thermometer). Anomalies are calculated as the difference between the mean value of a given period and the value



observed in 1809. Two different periods are used for comparison purposes: i) 1805-1808 of the SF1799-1813 dataset; ii) 1997-2021 of the SF1997-2021 dataset. Table 1 shows the temperature anomalies calculated for the year 1809.

As can be seen in Table 1, anomalies calculated with the period 1805-1808 show more negative values than the period 1997-2021, reflecting the fact that this later period is already influenced by global warming (IPCC, 2021). There is a clear difference between the February and March temperature anomaly in all cases, implying that there was an important drop in temperature. Also, the value of the April temperature anomaly is one of the lowest in all cases. Although May presents a higher value of the anomaly than April, the temperature anomaly in the months June and July decreases. The lowest value for the anomalies calculated with the period 1805-1808 is observed during the month of July.

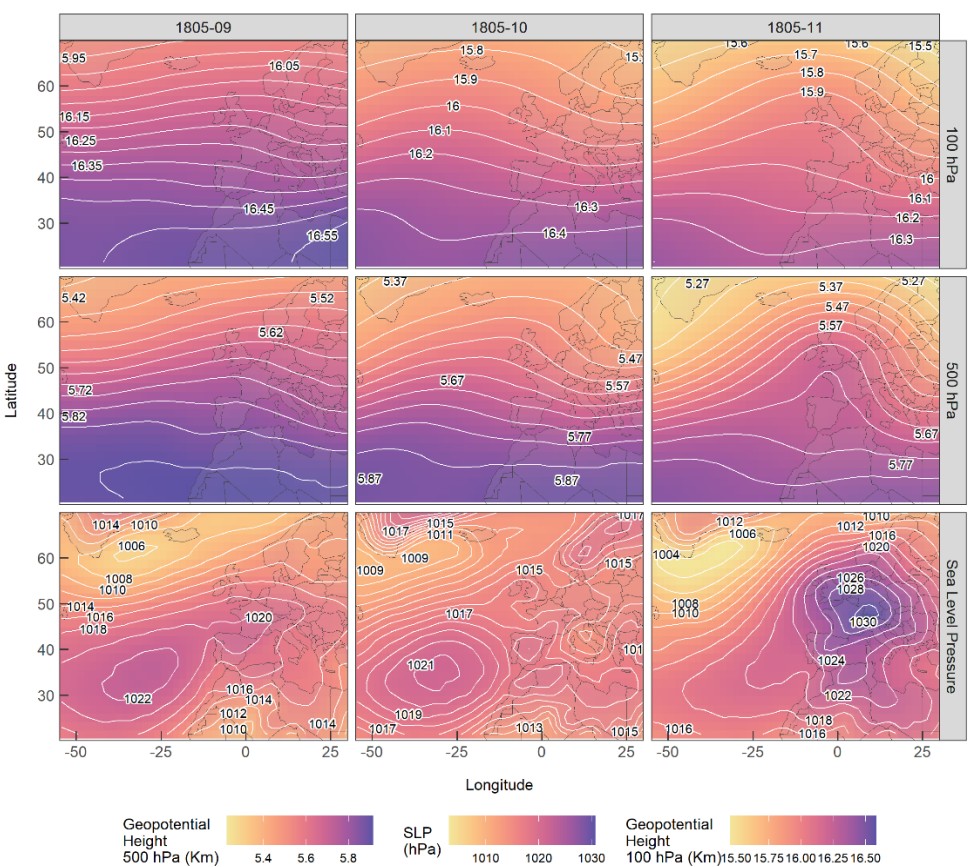


**Figure 10: SLP and geopotential height at 500 and 100 hPa of EKF400v2 dataset for the months of September to November 1805.**

Moreover, Fischer et al. (2007) analyzed precipitation data after 10 volcanic eruptions for the period 1769-2000 and found dry conditions in the winter of the eruption and the next winter in the IP. As can be seen in Fig. 8, there is a gap in precipitation in the summer months in all years, but the gap in the year 1809 is larger. Months that registered precipitation in 385 the year 1809 are January (82.82 mm), March (130.03 mm), November (39.09 mm) and December (36.96 mm). However, it is possible that the precipitation was not recorded due to some technical or human problems, but there is no information in





the metadata about this. The mean accumulated (anomalies) precipitation for the months of January, March, November and December of the SF1997-2021 (EKF400v2) dataset are: 58.6 (35.6), 56.8 (-20.5), 79 (-8.82), and 67 (-12.4) mm. Accumulated precipitation values for the months of January and March are higher than the mean accumulated precipitation

of the SF1997-2021 dataset. But, the precipitation of the last months of the year 1809 is less than the SF1997-2021 dataset. For the EKF400v2 dataset, more precipitation was estimated for the month of January, but less precipitation was estimated for the other three months. In addition, the pressure observations did not register any major drops in the year 1809 (as can be seen in the upper plot in Fig. 4).

**Table 1: Temperature anomalies in °C of the thermometer attached to the barometer and of the air temperature for the year 1809.**
**Anomalies are calculated using two comparison periods (1805-1808 and 1997-2021) of different datasets (SF1799-1813 and SF1997-2021, respectively).**

|  | 1805-1808 mean | | 1997-2021 mean | |
| --- | --- | --- | --- | --- |
|  | Thermometer barometer | Air thermometer | Thermometer barometer | Air thermometer |
| **January** | 2.23 | 5.54 | 4.35 | 4.62 |
| **February** | 2.32 | 4.72 | 4.44 | 4.26 |
| **March** | -0.07 | 0.80 | 1.43 | 0.38 |
| **April** | -1.21 | -0.77 | 0.10 | -0.97 |
| **May** | -0.46 | 0.47 | 1.18 | 1.42 |
| **June** | -1.22 | -1.06 | 0.88 | 0.87 |
| **July** | -2.28 | -2.64 | 1.14 | -0.32 |
| **August** | - | - | - | - |
| **September** | - | - | - | - |
| **October** | 0.55 | 2.54 | 3.27 | 2.50 |
| **November** | 0.31 | 2.22 | 2.70 | 1.95 |
| **December** | -0.15 | 0.77 | 1.69 | -0.72 |

Testimonies about fog, changes in the color of the solar disk or sunsets with intense colors after great volcanic eruptions can be found in the documentary sources (e.g.: (Guevara-Murua et al., 2014)). Haze skies can be observed in the summer months

in Spain due to the intrusion of dust from Africa (Russo et al., 2020) and the mentioned color changes can be observed. But haze skies are rarer to see in the winter months (Russo et al., 2020). The term "*calimoso*" (haze sky in English) is found in some days of January and February of the year 1809 in the description of the state of sky of the SF1799-1813 dataset. Specifically, the days are: 29/01-02/02, 07-08/02 and 27-28/02. The term "*calimoso*" also appears on March 21st. Thus, all these discussions suggest that the effects of the unknown eruption of 1809 could be observed in San Fernando.





## 6 Conclusions


The Cádiz-San Fernando region is one of the few places on our planet with a variety of overlapping series of early meteorological data. However, to the best of our knowledge, one of the longest and most interesting series (carried out by the staff of the Spanish Navy observatory) had not been recovered or studied until this work. In total, 45862 daily meteorological observations of the ROA dataset have been recovered for the period 1799-1813. The dataset is composed by

different meteorological variables such as atmospheric pressure, air temperature, precipitation or state of the sky. After applying a number of quality control steps a small amount of values (0.66 % of the data) were detected as suspicious in the digitization process and were corrected. Also, the units have been converted to modern units and the metadata has been described.

In the comparison between SF1799-1813 and SF1997-2021 datasets, a significant difference between the pressure values of

both series can be observed. Therefore, two adjustments are applied in different periods: for the period 1799-1801 and 1808-1813, a value of 10.3 hPa was added to the pressure observations of SF1799-1813 while for the period 1805-1809 a value of 4.9 hPa was added. Moreover, it has been added 0.22 inches of pressure to the original pressure observations of SF1799-1813 dataset for the period 1808-1809. The three datasets compared (SF1799-1813, SF1997-2021 and EKF400v2) show a similar behavior for the air temperature data and there is an excellent agreement between air temperature for the SF1799-

1813 and SF1997-2021 datasets.

The wind direction frequency analysis shows, when all months of the year are considered, five prevailing wind directions (NW, W, SW, SE, and ESE) for the period 1799-1813. This is in accordance with the location of San Fernando, which is closed to the Strait of Gibraltar. In general, the frequency of westerlies is higher than the easterlies.

The wind force analysis shows three predominant wind forces (*flojo* (Beaufort scale = 3), *fresco* (Beaufort scale = 6) and

*fresquito* (Beaufort scale = 5)) for the period 1799-1813. The predominant state of the sky is *claro* (clear sky) for the period 1805-1813.

The recovered precipitation data show great consistency with respect to the expected climatology: a seasonal cycle with almost no rainfall in the summer months. The years 1812 and 1813 recorded the maximum accumulated precipitation for the period 1805-1813. Unfortunately, observers only recorded precipitation readings when they occurred. Thus, the non-

registration of 0 mm of rain entails a certain indeterminacy in these data. In addition, we have found a pair of contradictory data between total monthly precipitation (0 mm) and number of rainy days (other than zero) in the months of February and March 1813.

Finally, we would like to highlight that the recovered data allow us to better understand the meteorological conditions of the Battle of Trafalgar in October 1805. The new available data indicates that the most plausible meteorological scenario is

characterized by the passage of a low-pressure system, contrary to other scenarios proposed in previous works. Besides the two application examples described here we would like to see this dataset used for different applications. We are confident that providing free access to this new historical dataset will foster such additional assessments by other researchers.



**Dataset access**

The SF1799-2013 dataset recovered is publicly available at the Zenodo open data repository at
https://doi.org/10.5281/zenodo.7104289.

**Author contribution**

N.B.-P. made the data analysis and wrote the first draft of the manuscript. M.C.G and J.M.V. located the original data and
initiated the collaboration. R.T., J.M.V. and M.C.G. wrote and revised the manuscript.

**Competing interests**

The authors declare that they have no conflict of interest.

**Acknowledgments**

This research was supported by the Economy and Infrastructure Counselling of the Junta of Extremadura through project
IB20080, and grant GR21080 (co-financed by the European Regional Development Fund). The research of N. Bravo-
Paredes has been supported by the predoctoral fellowship PRE2018-084897 from Agencia Estatal de Investigación
(Ministerio de Ciencia, Innovación y Universidades) of the Spanish Government.

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
