# Peer review of "Earliest meteorological readings in San Fernando (Cádiz, Spain)"

_Climate of the Past, 2022_

## Referee Comment (RC1)

I have revised manuscript entitled "Earliest meteorological readings in San Fernando (Cadiz,Spain)"

This paper recovered daily meteorological observation recorded at the Royal Observatory of the Spanish Navy during the period 1799-1813. The authors carried out important work of quality control and convert of units for the original data. Moreover, authors study meteorological condition during the Battle of Trafalgar and local effects of unknown volcanic eruption in 1809. Reviewer think that authors describe scientific results and conclusions presented in a clear, concise, and enough structured way. This work can contribute for understanding climate conditions during the 18th to the 19th centuries. However following minor revisions are needed before acceptance.

1. Title:   It is better to add analyze period of this study in title. (ex. " during the period 1799-1813") . It would be more attractive for the readers who are interested in climate in this period.

2. P4, l77: Authors mentions difference in observation time during the observation period. Although, authors carefully conducted quality control and convert of units, they did not mention to possible bias caused by changes in observation time in sub-daily observations. If there are possibility of bias, reviewer think it is better to mention it. Bias caused by changes in observation time during a day is documented in following paper in detail.

   Zaiki M, Kimura K, Mikami T. 2002. A statistical estimate of daily mean temperature derived from a limited number of daily observations. Geophysical Research Letters 29(18): 39.1-39.4.

3. Fig4: Authors compares pressure observations SF 1799-1813 series and monthly pressure average cycle of SF 1997-2021 series in the same figure. Reviver think it is confusing to draw pressure data for the different period (SF1799-1813) and (SF 1997-2021) within the same figure, because horizontal axis of this figure has only information for the period from 1799-1813.

4. Fig4. Author used EMKF400v2 dataset for the comparison. Reviewer think it is better to describe quality of this dataset around Iberian Peninsula.

5. Fig5. Authors display mean monthly air temperature series in this figure. Reviewer think it is better to add plots of each monthly values in this figure.

6. Figure 6: Label of wind direction is too small to distinguish.

7. P13, L287: Authors claim that "higher-than usual winter temperatures did occurred throughout Europe and possible even at the global scale". However, authors cited only studies for Switzerland and Spain. If authors claim "higher-than usual winter temperatures at the global scale", reviewer think it is better to cite more examples of high temperature anomaly in this period within and outside Europe.

8. P15 Fig9: If possible, reviewer think it is better to represent unit of precipitation by mm, because precipitation unit in Fig 8 is mm.

9. P16, L367: Authors mention that "temperature for the year 1809 of the SF1799-1813 dataset shows a decrease with higher(lower)than average during winter(summer) months (can be seen in Fig5). However, it is difficult to distinguish difference between summer and winter in Fig5. Moreover, if authors claim this seasonal difference in temperature anomalies caused by volcanic eruption., reviewer think more detailed explanations on mechanism are needed.

---

## Author Response (AR1)

REFEREE #1 comments.

**AUTHOR comments.**
* * *
I have revised manuscript entitled "Earliest meteorological readings in San Fernando (Cadiz,Spain)"

This paper recovered daily meteorological observation recorded at the Royal Observatory of the Spanish Navy during the period 1799-1813. The authors carried out important work of quality control and convert of units for the original data. Moreover, authors study meteorological condition during the Battle of Trafalgar and local effects of unknown volcanic eruption in 1809. Reviewer think that authors describe scientific results and conclusions presented in a clear, concise, and enough structured way. This work can contribute for understanding climate conditions during the 18th to the 19th centuries. However following minor revisions are needed before acceptance.

**Thank you very much for your comments. We are grateful that you appreciate the large amount of work we have carried out with quality control and converting the units, as well as the study on the weather conditions during the Battle of Trafalgar and the local effects of the unknown volcanic eruption in 1809.**

1. Title: It is better to add analyze period of this study in title. (ex. " during the period 1799-1813") . It would be more attractive for the readers who are interested in climate in this period.

**We agree with the referee's suggestion. Therefore, "1799-1813" will be added to the revised title of the manuscript.**

2. P4, l77: Authors mentions difference in observation time during the observation period. Although, authors carefully conducted quality control and convert of units, they did not mention to possible bias caused by changes in observation time in sub-daily observations. If there are possibility of bias, reviewer think it is better to mention it. Bias caused by changes in observation time during a day is documented in following paper in detail.

Zaiki M, Kimura K, Mikami T. 2002. A statistical estimate of daily mean temperature derived from a limited number of daily observations. Geophysical Research Letters 29(18): 39.1-39.4.

**The referee is right. Thus, the following sentence has been added to line 83: "Since there are differences in the observation hours from one period to another, it is possible that there are biases in the daily temperature caused by this fact (Zaiki et al., 2002)".**

1. Fig4: Authors compares pressure observations SF 1799-1813 series and monthly pressure average cycle of SF 1997-2021 series in the same figure. Reviver think it is confusing to draw pressure data for the different period (SF1799-1813) and (SF 1997-2021) within the same figure, because horizontal axis of this figure has only information for the period from 1799-1813.

**The purpose of the black continuous line in Figure 4 is to show a repeating climatological cycle for comparison purposes with the recovered pressure data at the monthly scale during the 1799 -1813. The black line shows the monthly mean values for the 25 years of available data,**

**i.e. the SF1997-2021 dataset, that is, the mean of each month of all the years in that period because the intention is to present a climatological normal to do a comparison. This is a very standard procedure to compare all old data observations with a well-known seasonal cycle for present climate. See, for example, Figure 6 of the study by Rodrigo and Millán (Early meteorological observations in West Africa during the 18th century, International Journal of Climatology, vol. 42, issue 16, 9753-9766, 2022) or Figure 2 of the article by Trigo, Vaquero and Stothers (Witnessing the impact of the 1783–1784 Laki eruption in the Southern Hemisphere, Climatic Change, vol. 99, 535–546, 2010). Therefore, we do not think it is confusing to represent a climatology from this dataset with the other two datasets, even though the horizontal axis is only relative to the period 1799-1813.**

1. Author used EMKF400v2 dataset for the comparison. Reviewer think it is better to describe quality of this dataset around Iberian Peninsula.

**We acknowledge the reviewer's suggestion but would like to underline that such exercise has been performed in depth by other authors. Please, see the article by Valler et al. (2021). You can see Figure 3 of this article to check the quality of the dataset on the Iberian Peninsula.**

1. Authors display mean monthly air temperature series in this figure. Reviewer think it is better to add plots of each monthly values in this figure.

**We suppose that you are commenting our Figure 5. It is true that the presence of the annual cycle in Figure 5 means that many details of possible interest are lost. We have done the proposed exercise of representing the monthly values in each plot. Thus, we have obtained 12 plots (one for each month). Fortunately, no other details of interest have appeared and only the anomalies already previously indicated by us are highlighted. For this reason, we have decided not to incorporate the 12 plots into the new version of the manuscript.**

1. Figure 6: Label of wind direction is too small to distinguish.

**The wind direction label has been changed to better distinguish the label.**

1. P13, L287: Authors claim that "higher-than usual winter temperatures did occurred throughout Europe and possible even at the global scale". However, authors cited only studies for Switzerland and Spain. If authors claim "higher-than usual winter temperatures at the global scale", reviewer think it is better to cite more examples of high temperature anomaly in this period within and outside Europe.

**Thank you very much for the comment. We agree with you and the text has been changed as follows: "Therefore, it seems that this higher-than usual winter temperatures did occurred throughout Southwest Europe".**

1. P15 Fig9: If possible, reviewer think it is better to represent unit of precipitation by mm, because precipitation unit in Fig 8 is mm.

**The unit of precipitation has been changed to mm in Fig. 9.**

2. P16, L367: Authors mention that "temperature for the year 1809 of the SF1799-1813 dataset shows a decrease with higher(lower)than average during winter(summer) months (can be seen in Fig5). However, it is difficult to distinguish difference between summer and winter in Fig5. Moreover, if authors claim this seasonal difference in temperature anomalies caused by volcanic eruption., reviewer think more detailed explanations on mechanism are needed.

We understand the reviewer's difficult that may result from the use of several lines and their different behavior over time. Therefore, we have now clearly marked the year 1809 with two vertical thin lines identifying the beginning and ending of that year. We hope this facilitates the visualization of the daily air temperature (grey line) staying above the winter average for that year and below the summer temperatures observed in the previous years, i.e during the summers of 1805, 1806, 1807 and 1808.

In respect to the mechanism related with volcanic eruption and climate, we think that some review articles should be cited in the manuscript. As an example, we now cite the classical review by Robock, A. (2000), Volcanic eruptions and climate, Rev. Geophys., 38( 2), 191– 219, doi:10.1029/1998RG000054.

REFEREE #2 comments.

**AUTHOR comments.**
* * *
Reviewer comment on _Earlyiest meteorological readings in San Fernando (Cádiz, Spain)_ by Nieves Bravo-Paredes et al.

I have read the paper by Bravo-Paredes et al and I think that there are parts of the paper which are valuable and, therefore, I am inclined to support its publication. On the other hand, I find that some parts of the paper are extremely speculative and, therefore, I suggest some major revisions before the paper may be accepted in Climates of the Past.

**Thank you very much for all these comments.**

Major points

1. In general, the paper needs a serious revision of English. I have seen that there is a comment suggesting many changes before. Even though English is not my mother tongue, I have found many examples of sentences hard to read, with lack of correct position of words, lack of proper conjunction of sentences and similar errors. For instance, the junction between sentences in lines 15-17 of the abstract is not correct. There are many such instances. This must be corrected before a new version is submitted.

**Thank you very much for this comment. An English revision has been done as suggested by the referee.**

2. In page 11, the authors discuss a result which is a little bit surprising for me, which is the fact that summer temperature (page 11) during 1799-1813 was apparently lower than current ones (1997-2021). This is a little bit surprising for me. The authors mention later that (sic) this higher than usual winter temperatures. I, therefore, think the authors must clarify whether they mention winter or summer.

**On lines 274-275 the following is written: "The temperature of SF1799-1813 in summer months is slightly higher than the monthly average of the SF1997-2021[…]". This sentence is related to the period 1805-1809.**

**On lines 277-279 the following is written: "The temperature values for the first months of 1809 of SF1799-1813 dataset are considerably higher than the monthly temperature of the SF1997-2021 and EKF400v2 datasets for these months". This sentence is related only to the first months of the year 1809 (i.e winter months).**

**Therefore, in summary (i) the summer temperatures for the period 1805-1809 are not lower than the monthly average of the SF1997-2021 and (ii) the temperature values for the first months of 1809 (i.e winter months) are higher than the corresponding monthly temperature averages of the SF1997-2021. We have slightly revised our text to make these points clearer to the reader. On lines 274-275 we have written: "The temperature of SF1799-1813 in summer months (for the period 1805-1809)[…]"**

On the other hand, they select some locations over Europe (Madrid, Mallorca San Fernando or Switzerland) that show a similar behaviour, with some of them (Switzerland) really far away. I think the authors must make a better effort in this part either by a deeper analysis of literature or additional reconstructions (instrumental or multi-proxy) covering that period, so that they can give a satisfactory explanation here. Otherwise, for me, it is hard to accept their line 286-287 without further (and better) analysis.

**We do not select locations over Europe. We looked for all available instrumental temperature data in Spain that covered the year 1809 to check the behavior of the first months of the year 1809. Moreover, we found the Switzerland series and the comparison was interesting. Note that it is very difficult to find early meteorological measurements and we only found available these temperature series. The Lisbon series for example only starts in 1815. In any case, we will review the text of lines 286-287 and underline that we have made an effort to compare with all available Temperature data for that period.**

3. Page 14, line 320. Authors write (sic) the highest precipitation values were recorded in these months and they previously mention Nov 1812, Dec 1812, Oct 1813, Nov 1813 and Dec 1813 (line 319 and Figure 9). However, when looking to Figure 8, if the label of the horizontal tick marks corresponds to January, I don't see that November 1812 or November 1813 are particularly rainy at all. This is consistent with the maps in Figure 9. Thus, I wonder whether the sentence is true, or whether they refer to anomalies or whether there is an alternative explanation for their sentence. If there is not a better explanation, I find the sentence misleading and not consistent with the information provided by the paper.

**We are sorry for any confusion. We changed the sentence in line 321 as follows to avoid misunderstandings: "The highest precipitation values were recorded in the months of December 1812 and October and December 1813 for the period 1805-1813 (see Fig. 8)".**

4. My major objections, however, deal with the application of the dataset for its application to two case-studies, the Battle of Trafalgar and the volcanic eruption in 1809. I find this part is in general very speculative and that the authors do not provide sound support for their interpretations.

1. The Battle of Trafalgar. In this case the interpretation seems to me relatively straightforward, but I find that this interpretation is actually sound because of the use of the EKF400v2 dataset rather than because of the analysis of a single time series in San Fernando. I can accept that this part seems reasonable, and that the dataset prepared by the authors points in the same direction as the synoptic data reconstructed by EKF400v2.
   **We understand that the two case studies are the weakest part of our study. We tried to offer a couple of "applications" of the recovered data. But we acknowledge that it is difficult to do something meaningful using the data from a single location (San Fernando).**

2. Volcanic eruption of 1809. I find this part really speculative.
• On the one hand, the correct date that the eruption happened can not be accurately fixed. The authors find the lowest anomalies during July (at San Fernando). Being the

volcanic influence on temperature probably global, I find urgent that the authors cross-check their findings with global or hemispheric reconstructions. If this is true in San Fernando and not an artefact, it shuld be true for a wide area of Europe as well. Is it?

- I find strange the phrase anomalies calculated with the period 1805...2021) in lines 373-174. I would say that the anomalies computed with the series $x$ by using its own average $\bar{x}$ ($\Delta x^* =x-\bar{x}$)  will always be higher (not lower) than the ones calculating after adding a constant positive bias (greenhouse warming) to the series ($\Delta x^\dag =x-(\bar{x}+\bar{X}_{GH})$). Thus, I can not understand this sentence by the authors. On the other hand, this must be reconciled with point 2 above, which deals with the fact that the authors mention that current temperatures are lower than the ones in early XIXth century (which I still find hard to believe, but which is opposite to their assertion here).

- I find sentences 381-392 totally speculative and I think they are not supported by a detailed analysis.
  **We again understand that the two case studies are the weakest part of our study and, in particular, the part related to volcanic eruption of 1809 is really speculative. Therefore, we agree that, in this case, the best option is to simply remove it.**

Thus, the references to the volcanic eruption are, in my opinion, a problem of this paper. I think that the production of the dataset and making it available for other scientists has some value that might merit a publication. However, the analysis of the volcanic eruption is a very weak point of the paper. The authors should make it robust or just remove it.

**Thank you very much for your encouraging comment. We also think that the production of the dataset and making it available for other scientists has some value that might merit a publication. Therefore, as mentioned above, we will remove this part in the new version of our manuscript.**

Minor points

1. I guess words tiempo, planetas, longitudes terrestres and fisica celeste in line 69 should be translated to English for better multicultural understanding
   **Since the original name is in Spanish, we think it is better to writhe it in Spanish. In any case, we will write in parentheses the English translation.**

2. Line 76. The authors mention subdaily observations but then, later, they mention that data collection was carried out at 2 pm (which is just daily from my POV).
   **We agree with the reviewer that this information was not sufficiently clear. In the revised version of the manuscript, lines 74-76 are changed as follows: "Pink and green bars correspond to sub-daily observations: 6am, 12am, 6pm and 12pm hours in pink; 8am and 2pm hours in green. Blue bars correspond to daily observations measured at 2pm".**

3. Line 116. "The prevailing wind direction...1813". I think the authors must expand this explanations. Was any recording instrument involved? Was that recording just evaluated subjectively by observers?
   **Unfortunately, we only know what is written in the metadata provided by the handwritten source: "the most prevalent address in the 24 hours of the day". We have added to the text: "[…], although there is no further indication in the original manuscript".**

4. There are some missing references in page 8.
   **This problem has already been solved.**

5. Is there any good reason that a relatively truncated approach to local gravity (Eq 7) is used instead of current values of gravity at the area (which probably are better and more precise)?
   **We are following the expressions recommended by the WMO. In any case, we do not know the values of local gravity for such early years.**

6. Page 9. Together with the value of $R^2$, the sample length and, even better, the confidence interval of the null hypothesis (do thermometers read linearly related temperatures?, I guess).
   **We will change our manuscript according to this comment.**

7. In Figure 4 (low panel) and Figure 5, I find problematic the selection of the solid data for the modern period and the dashed lines for the data recovered by the authors. The reason is that the modern period is very far in time and, as such, it is expected to be less related to the actual measurements than the means computed from the daily data. However, the use of solid color for the modern dataset makes it more visible. Thus, I really encourage the authors to use solid line for the monthly (daily) data retrieved by the authors in this study and dashed lines for the data from the modern 2006-2021 period. The text mentioning these two figures (mentions to solid or dashed lines) should be changed accordingly.
   **The colors of Figures 4 and 5 have been changed as suggested by the referee.**

8. Line 296. Authors mention that westerlies are more common in the area, which is basically to be expected, since it is located in extratropical latitudes of the Northern Hemisphere. May be just add a sentence explaining this. For me, it is not a surprise.
   **We have added: "In general, the frequency of westerlies is higher than the easterlies, as it is expected (Sousa, 1987)".**

COMMUNITY #1 comments. Michael Chenoweth

**AUTHOR comments.**

A number of minor changes are offered here to improve the grammar in the English text:

Line 7 - change download to "downloaded"

Line 9 - insert the word "are" between "previously more" to read "previously are more"

Line 12 - change "composed by different" to "composed of different"

Line 15 - change "data have been described" to "data are described"

Lines 22-23 - change

Line 36 - change "from 18th century" to "from the 18th century"

Line 36 - change "focused in 19th" to "focused on 19th"

Line 62 - change  "San Fernando in the Bay of Cadiz" to "San Fernando on the Bay of Cadiz"

**Thank you very much for these changes that will improve the grammar and English style of the text of the revised manuscript. Naturally, all these changes were made as suggested by the referee.**

Line 69 - Reword the sentence to read: "The data compiled in SF1799-1813 came from data generated by a plan to carry out astronmical....."

**The sentence has been rewritten as follows: "The ROA dataset used here (hereafter SF1799-1813) was obtained under the scope of a larger plan to increase astronomical observations at the Observatory starting in 1798".**

Lines 74 & 75 - Change the first two sentences to: "The ROA observations were made using different instruments and different units. A total of 45862 discrete observations were made from 1799 to 1813."

Line 81 - change "provokes" to "means"

Line 92 - change "There is not station elevation..." to "There is no station elevation..."

**All these three proposed changes were made as suggested by the referee.**

Line 94 - change "100 feet from sea level" to "100 feet above sea level". Is this Burgos feet or unknown?

**The change was made as suggested by the referee. Unfortunately, the text of the manuscript does not specify the type of foot corresponding to this measure. In any case, since the height belongs to an area close to the sea, the use of a Burgos foot or an English foot (with a relative error between them that can be estimated at 2%) does not entail absolute errors of a large magnitude (of the order of 2 meters in height).**

Lines 143-144 change "instead of the real number of the original document" to "instead of the actual number in the original document"

Line 236 - Insert the word "the" ahead of each instance of "dashed gray line" .... "solid gray line"

Line 238 - THE black line ...

Line 238-239 - THE dashed black line

Line 239 - This LINE corresponds to the

Lines 261-262 - as above, insert the word "the". THE daily average air temperature for 1997-2021 is depicted by the solid black line. THE dashed black....

Line 263 - change "Just like pressure data" to "Just like the pressure data"

Lines 308-309 - change "months there is not rainfall data" to "months there is no rainfall data"

**Thank you very much for calling our attention to these errors. All changes were performed according to the referee's suggestions.**

Other comments:

Lines 20-23 - The authors discuss the importance of the creation of long-time series of meteorological variables. However, as this paper does not have continuous data to append to a Cadiz-centered series then I do not think this is pertinent to the discussion. Perhaps a focus on data search and rescue that can direct searches for other data that might fill the remaining gap.

**Thank you very much for this comment. Our purpose in those first sentences of the manuscript was to introduce the reader to the literature on early instrumental weather observations, regardless of their continuity and discontinuity. In any case, we should mention that most of the early instrumental records are relatively short series, often even shorter than the one presented here (as shown in Table 1). Thus, a collection of this kind of time series relating to the same place, or different places with similar climatic characteristics, can be extremely useful for better knowing the climate of the past. In this sense, we have rewritten some parts of the introduction, following the advice of the referee.**

Lines 94-96 - I do not agree with the reasoning for the choice. The same instruments could just have easily been moved to another site. It is best to leave this unknown or without reasonable uncertainty bounds.

**Thank you very much for this comment. We are aware that this is an arbitrary choice. However, when considering the raw data, we observe that the average pressure values for the first third periods (both obtained with the George Adams barometer) are practically the same. Therefore, the assumption of a stationary height seems quite reasonable. In any case, we have added a sentence explaining our rational.**

Line 99 - do we have any reason to believe it is the same George Adams barometer in both periods?

**In the metadata, it is only written that the barometer is a George Adams barometer in both periods. As we have previously explained, when analyzing the raw data, we can check that the average pressure values of the first period and the third period (both obtained with "the" George Adams barometer) are very similar. Thus, we are confident that the barometer used is the same, which is also the simplest assumption. In any case, we have modified the previous sentence to inform the reader of these facts.**

Lines 113-114 - since this is an English language paper, it is better to convert the WpSW to WxSW and so forth, while noting it is "p" in Spanish. That is, reverse the usage as presently presented.

**The terms of wind direction were converted as suggested by the referee.**

Line 117 - what level of precision were the prevailing wind data given in 1812-13?

**In the metadata it is only written that the wind direction data was recorded as the predominant direction in 24h. Unfortunately, there is no more information, and we cannot offer a clear response on the precision level of this wind data.**

Lines 119-121 - How were the wind force terms converted into a numerical value?

**The wind force terms have not been converted to numerical values. We stablish in line 119: "Wind force observations were recorded subdaily in textual form using descriptor".**

**In any case, we comment that the archaic terms associated with the different categories of Beaufort description could be written in many possible ways, e.g., the term "bonancible", in English "moderate breeze" (Beaufort number = 4), can be written in Spanish in the following ways: "abonanzado", "apacible", "apaciguado", "benigno", "bonancible", "bonanza", etc. García-Herrera et al. (2003) wrote a dictionary to express archaic wind force terms in a comprehensible form to the modern-day reader. In addition, they wrote the equivalent Beaufort force number for each term. This dictionary has been used to identify archaic terms that appear in the data retrieved with the Beaufort description. Once the term was identified (e.g., "bonanza"), it was converted to the Beaufort description ("bonancible" – Beaufort number = 4).**

**Finally, we must apologize because the reference provided on this topic is not the correct one. Instead of Calvo et al. (2008), the correct reference should be García-Herrera et al. (2003). It has been changed in the revised manuscript.**

Line 122 - Are the wind force terms in 1812-13 consistent with the earlier series?

**The names used to characterize the wind force terms did not change throughout the whole period of the series. Therefore, the wind force terms are consistent with the previous periods.**

Lines 139-141 - if the basic quality control steps are identical to those done with other records, a reference to the source could be included here.

**The steps of this quality control are not entirely identical to those provided by others, although they are similar. In any case, in regard to this issue, a reference has been included: Vaquero, J.M., Bravo-Paredes, N., Obregón, M.A., Carrasco, V.M.S., Valente, M.A., Trigo, R.M., et al (2022) Recovery of early meteorological records from Extremadura region (SW Iberia): The 'CliPastExtrem' (v1.0) database. Geoscience Data Journal, 9, 207– 220. https://doi.org/10.1002/gdj3.131**

Lines 143-148 - I think this section could be removed without losing any information as to the checks described after these lines.

**This paragraph gives a brief introduction to typical errors that can be found when digitizing data. Some readers are not aware of these digitation caveats. Therefore, we think it is better not to remove it.**

Line 182 - can remove "The density of the mercury changes with temperature." This is redundant as the other sentences say the same.

**We agree with the referee and so the sentence has been removed.**

Lines 190-191 and 198 - There are two bold-faced notes that begin with Error. The note is in Spanish. If the authors need to provide additional information then this should be done as I am not sure what their intention is concerning the notes.

**We are sorry for any confusion this error may have caused. It is a Word error for having a link of the figures, tables and equations with the text (cross references). In this case, this connection to the equation and the text must have been lost and caused the error. It has already been solved.**

Line 193 - change Pascal to hectoPascal

**It has been changed.**

Line 210 - change "the coefficient of determination" to "the correlation coefficient between"

**The coefficients of determination and correlation are not the same. In this work, we have used the coefficient of determination. We are using the conventional name for the coefficient of determination $R^2$, which is the square of the correlation coefficient.**

Line 225 - how were the adjustments made to compare them - were the same times of observation available in both periods? Were these adjustments only made to monthly average values or to the original readings?

**Please, see our next response.**

Lines 235-239 - the usefulness of this figure hinges on the questions concerning line 225.

**We have not performed a least squares adjustment here. We are only comparing the average values of the different periods to better assess the systematic deviations between the readings of the different periods. We will try to better explain our procedure in the new version of the manuscript.**

Lines 249-250 - change "Due to this fact, it has been added 0.22 inches of pressure to the original pressure observations" to "Due to this UNDER-READING, 0.22 inches ARE ADDED to the original pressure observations"

**It has been changed as suggested by the referee.**

Figure 4 caption - you need to state what the black line is, which you do in the text.

**The description of the solid and dashed black lines are written in the caption. Please note that, in the preprint publications process, the caption of Figure 4 has been cut and appears at the top of the next page.**

Lines 256-259 - Given the different observation times, and the lack of details on the correction procedure with respect to these differences, I am not clear on why I should believe the data shows similar "behavior", by which I think the authors mean the average annual, monthly, and diurnal ranges of pressure. The gridded data may or may not be accurate but I have no way of comparing them based on the information as presently provided.

**We agree with the reviewer and we have re-written this sentence according to your comments.**

Lines 261-262 - Why is the DAILY for 1997-2021 being compared to the MONTHLY for 1805-1809? Why not MONTHLY for 1997-2021 to compare with the MONTHLY gridded 1805-1809 data?

**We are not making a quantitative comparison of daily data with monthly data. We are simply making a figure that contains daily and monthly data for better visualization. The daily and monthly data have been plotted together because it is easier to see if the behavior of the daily values is far from the mean of the modern or reanalysis values. This is a standard procedure to compare all old data observations with a well-known seasonal cycle for present climate. See, for example, Figure 6 of the study by Rodrigo and Millán (Early meteorological observations in West Africa during the 18th century, International Journal of Climatology, vol. 42, issue 16, 9753-9766, 2022) or Figure 2 of the article by Trigo, Vaquero and Stothers (Witnessing the impact of the 1783–1784 Laki eruption in the Southern Hemisphere, Climatic Change, vol. 99, 535–546, 2010).**

Line 270 - change coefficient of determination to CORRELATION COEFFICIENT and elsewhere where it appears after this line. Monthly values for all three series need to be compared, not the mix of monthly and daily averages.

**See previous answer of line 210.**

**The monthly values of the three series have been compared. Daily and monthly values have not been mixed. To avoid misunderstandings, it will be better specified in the text that the comparison is obtained at the monthly scale.**

Lines 274-275 The average temperature in the summer months in 1799-1813 is slightly higher than the same months in 1997-2021. Does this likely mean that thermometer exposure in 1799-1813 is biasing the readings too high?

**We agree that this could constitute a possibility. Nevertheless, please notice that the average temperature in the summer months of the independent EKF400v2 dataset is quite similar to those of SF1799-1813 dataset.**

Lines 275-276 - "and is quite similar to the monthly temperature of EKF400v2 dataset." Make clear if this is the same summer months. If not, state which months are being compared and separate it from the previous sentence if not just the summer months.

**To avoid misunderstanding, the sentence has been changed as follows: "The temperature of SF1799-1813 in summer months is slightly higher than the monthly average of the SF1997-2021 dataset, while being quite similar to the monthly average of the EKF400v2 dataset."**

Lines 276-288 - The most direct comparison is presumably between the gridded data point nearest to Cadiz/San Fernando and there is a significant difference. I think a more direct way would be to look at the prevailing winds and see if they support sustained - presumably southerly wind components - during these months. If so, then this might support the higher temperatures and suggest an under-reading in the gridded series. If not, then there is a possibility of an issue with the data at the observatory at this time.

**The prevailing wind directions in these months (January, February and March) are SW, SE, W and S.**

Line 285 - omit the words "of the world" at the start of the line.

**The words were omitted.**

Lines 287-288 - occurred throughout SOUTHWEST Europe. The remainder of the lines can be then eliminated as there is no reason to speculate about a supposed global scale warming in a winter that followed a major volcanic eruption in late 1808.

**Thank you very much. We have included the word "Southwest" and we have removed the remainder of the line, eliminating this speculative idea.**

Lines 289-294 - how do these winds compare with modern data. A better use of the wind direction and wind force data would be to calculate, however crude, a scalar wind (zonal and meridional components). Deviations of these components from the average could be used to see if they were consistent with estimated temperature anomalies. As presently written, there is little useful information to be gleaned from this paragraph.

**Wind direction data have not been compared with modern data.**

Lines 308-309 - Is this actually saying that no rainfall was recorded in these months? Another question: how frequent is snowfall at the site? It may be that precipitation can simply be called rainfall if snow is rarely observed or measured.

**Precipitation data appears in the manuscript, for example (the amount of precipitation presented is not real), as follows: January 2p 5l, February 1p 4l, March 2p 8l, April 3p 9l, May 1p 0.5l, October 2p 1l, November 2p 5l, December 1p 3l.**

**These sentences explain the absence of those months in the precipitation record. It may be that it was not recorded because there was 0 mm of precipitation or because of the absence of observers.**

**Note that, to the best of our knowledge, the last time it snowed in Cádiz was in 1954. Therefore, it is very rare that it snows in San Fernando.**

Lines 309-312 - Just because the pressure falls, does not mean rainfall will occur. I think that given the frequency of rainfall observations, weekly readings only, and otherwise only monthly totals, then the check against daily pressure is not very meaningful, particularly in the summer half of the year. Remove the sentence "According to the represented...data are consistent."

**We understand the reviewer's comment ("just because the pressure falls, does not mean rainfall will occur"). In any case, in SW Spain rainfall most likely occurs when you observe a significant drop in the atmospheric pressure. Therefore, we are using this fact to perform a "double checking" that can be useful to detect systematic errors in the measurements.**

Lines 321-323 - There is definitely a difference between the unrecorded, or lost, measurements and the rain days. Remove "Therefore, there could be some error in these months."

**If there are rainy days recorded and the amount of accumulated precipitation does not appear, then there may be an error in either the rainy days or in the accumulated precipitation. Therefore, we think it is better not to delete it but we have added to the sentence: "[…], either in the identification of the rainy days or in the amount of accumulated precipitation".**

Lines 326-328 - Are the gridded values of pressure anomalously low or not in the wet months? A climatology for each month would be useful to compare.

**As can be seen in Fig. 4, where the pressure values of the EKF400v2 data set are represented, the average of the pressure values is around 1020 hPa, so the pressure values shown in these sentences are not anomalously low.**

Line 339 - can remove "(but highly cited)" and make it "the circa 1809 unlocated volcanic eruption."

**Changed accordingly.**

Lines 340-361. There is no reason to use the monthly EKF400v2 dataset to try and compare with daily data for the battle. This gives no useful information to compare with what is known from the surface data in Wheeler (1985). The daily data from the observatory is consistent, as the authors state, with the lower pressure and rains noted but that is all. The best way to study the weather at Trafalgar is to map out a larger region at the sub-daily scale using land data and ship observations.

**We agree with the reviewer-s comments. In fact, taking into account these comments and also following the recommendation of referee #2,  we opted to remove this part of the manuscript related to Trafalgar Battle that is too speculative.**

Lines 363-373. The reduced temperature range in the winter of 1809, which was apparently unusually warm in the area, might simply be due to some combination of warm southerly winds, enhanced cloud cover, and perhaps higher wind speeds (which the authors could check from the data) and not invoke the volcano. The summer anomalies are more interesting, because one would expect clearer skies to prevail in the summer half of the year and the anomalies for 1805-08 are consistent with cooler temperatures, possibly associated with an aerosol layer diminishing incoming solar radiation. There is no reason to use 1997-2021 in the comparison and I would drop this part of the discussion and remove it from Table 1.

**In the sentence "The temperature for the year 1809 of the SF1799-1813 dataset shows a decreased range with higher (lower) than average temperatures during winter (summer) months" is described the temperature for the year 1809. And in the next sentence "This decrease range could be partially caused by the unknown volcanic eruption of 1809 that likely induced cooler summer temperatures" it is emphasized the PARTIAL participation of the volcanic eruption that probably induced cooler summer temperatures. Therefore, no reference is made to the apparently warmer temperatures of early 1809.**

**In any case, as mentioned when answering the previous question, we opted to remove this part of the text in the new version of our manuscript.**

Lines 382-390 - there is too much speculation about alleged gaps in observed rainfall in 1809 to make any useful comments relative to the cited literature. If the rainfall was above average as stated in January and March 1809, that is consistent with my comments about a warm and wet

winter reducing the diurnal range of temperature. The authors need to look at all of their data to see if there is internal consistency between the data and what they assert is occuring or might be occuring.

**We agree and will remove this part of the text in the new version of our manuscript.**

Lines 398-404 - This is a more interesting description of possible aerosol effects on the state of the sky. Is there any information on such sky states in later eruptions such as Krakatoa (1883)?

**In relation to the eruption of Krakatoa, it can be read in Kondratyev (1984): "[…] blue and green sun were observed for weeks and red sunsets persisted for up to three years; the volcanic dust, shielding out solar radiation, is believed to have lowered the global surface temperatures in the following few years […]". Therefore, although it is scarce, there is information about the state of the sky from other eruptions.**

**Kondratyev, K. (1984). Volcanoes and Climate. In World Climate Research Programme (nº 166; World Climate Paper-54).**

Line 405 - I think the first sentence overstates the rarity of such records, particuarly in Europe. More useful, is the fact that it is on the periphery of Western Europe far from many other such sites in Europe.

**We agree and will remove this part of the text in the new version of our manuscript.**

The authors need to note that the data are not directly homogeneous or part of a continuous time series, so its usefulness is limited until and unless other data and/or methodologies can allow the construction of such a time series. Some internal checks on some of the subsets of data might be useful - compare day versus night diurnal ranges of temperature in 1799-1801 with modern data to see if unusual patterns exist, which might reveal information about the exposure of the thermometer. This might lead to a more accurate way to adjust these isolated series.

**We agree with the referee. In fact, this is what we have tried to stress in the introduction and discussion sections of our manuscript. Please, see the changes that we have made in the new version of our manuscript.**

Lines 433-436 - The understanding of the weather of the battle of Trafalgar was not really significantly improved. The data is broadly consistent with what is known but the single record itself cannot do this - more data over a larger region is required as mentioned earlier.

**We agree and will remove this part of the text in the new version of our manuscript.**